# Simultaneous Gut-Brain Electrophysiology Shows Cognition and Satiety Specific Coupling

**DOI:** 10.3390/s22239242

**Published:** 2022-11-28

**Authors:** Pragathi Priyadharsini Balasubramani, Anuja Walke, Gillian Grennan, Andrew Perley, Suzanna Purpura, Dhakshin Ramanathan, Todd P. Coleman, Jyoti Mishra

**Affiliations:** 1Neural Engineering and Translation Labs (NEATLabs), Department of Psychiatry, University of California, San Diego, CA 92093, USA; 2Department of Bioengineering, University of California, San Diego, CA 92093, USA; 3Department of Mental Health, VA San Diego Medical Center, San Diego, CA 92108, USA; 4Center of Excellence for Stress and Mental Health, VA San Diego Medical Center, San Diego, CA 92108, USA

**Keywords:** EEG, electrogastrography, resting state, working memory, phase-amplitude coupling

## Abstract

Recent studies, using high resolution magnetoencephalography (MEG) and electrogastrography (EGG), have shown that during resting state, rhythmic gastric physiological signals are linked with cortical brain oscillations. Yet, gut-brain coupling has not been investigated with electroencephalography (EEG) during cognitive brain engagement or during hunger-related gut engagement. In this study in 14 young adults (7 females, mean ± SD age 25.71 ± 8.32 years), we study gut-brain coupling using simultaneous EEG and EGG during hunger and satiety states measured in separate visits, and compare responses both while resting as well as during a cognitively demanding working memory task. We find that EGG-EEG phase-amplitude coupling (PAC) differs based on both satiety state and cognitive effort, with greater PAC modulation observed in the resting state relative to working memory. We find a significant interaction between gut satiation levels and cognitive states in the left fronto-central brain region, with larger cognitive demand based differences in the hunger state. Furthermore, strength of PAC correlated with behavioral performance during the working memory task. Altogether, these results highlight the role of gut-brain interactions in cognition and demonstrate the feasibility of these recordings using scalable sensors.

## 1. Introduction

Recent studies have illuminated electrophysiological coupling between the gut and the brain. Gut-brain bidirectional interactions are suggested to occur via a number of distinct processes that includes enteroendocrine cell signaling, immune signaling [1], chemical communication via neurotransmitters, microbiome signaling [2] and mechanical transduction of stomach stretch receptors with afferents to the central nervous system [3]. In this study, we were specifically focused on the electrical communication between the gut and the brain [4]. Communication can be descending, in which the brain imparts directions to the stomach, or ascending, in which the stomach transmits information to the brain. One study in rats showed that parasympathetic control of the stomach originates largely from the insula, medial prefrontal cortex, and the areas associated with interoception and emotional processing, and the sympathetic control largely from areas such as the primary motor, sensorimotor cortices, secondary motor cortices [5]. Rebollo et al. described a gastric network in which the amplitude of resting state brain activity, measured with magnetoencephalography (MEG), is coupled to the phase of the gastric basal rhythm measured with electrogastrography (EGG). This gastric network consisted of sensorimotor brain regions including the right primary somatosensory cortex (SIr), bilateral secondary somatosensory cortices (SII), medial wall motor regions (MWM), portions of the parietal and occipital cortex, precuneus and the cingulum [6,7]. Overall, the brain regions involved with the gastric network map bodily space through touch, action, and vision, and are thus hypothesized to be linked with sensorimotor aspects of feeding.

In this study, we sought to answer several key questions about the electrophysiological coupling between the gut and the brain. First, we wanted to understand whether electroencephalography (EEG) is a sensitive tool for measuring the neural aspects of gut-brain coupling. To the best of our knowledge, there has not been a systematic investigation of gut-brain coupling using EEG-derived brain measures. As EEG is a far more inexpensive and scalable tool for measuring brain activity than MEG, a demonstration of EEG and EGG coupling could stimulate new research in this field. Further, in our review of the literature, we note that electrophysiologic gut-brain coupling has not been explored outside of the resting-state condition [4,8]. Hence, we were interested in studying whether similar coupling can be observed during cognitive states. Finally, there have not been any EGG based studies investigating how gut-brain physiological coupling during cognition differs based on hunger vs. satiety.

We use simultaneous EEG and EGG to acquire the signals from the brain and the gut, and study whether we can observe coupling between the electrophysiological oscillations in both systems. While brain oscillations are widely studied in the delta to gamma band [1–40 Hz] range [4,9,10,11,12,13,14], the gastric oscillations undergo a relatively slow rhythm (0.05 Hz) sourced from the intrinsic pacemaker Cajal cells lining the digestive system tract [8,15,16,17]. Here, we simultaneously studied electrophysiological activity from both the brain and the gut during both the resting state and a working memory task, across both hunger and satiety states. Our earlier studies have suggested that energy metabolic homeostasis is a core functional property of neuro-modulatory systems [18] that controls cognitive effort and engagement. We chose a working memory task since the state of hunger has been shown to have a significant influence on working memory performance in humans [19,20], and cognitive effort during working memory may be influenced by the major energy source in the gut and its interaction with brain processes [21,22]. Based on prior studies with MEG, we examined phase-amplitude coupling in which the phase of a low-frequency (gut) signal modulates the amplitude of a high-frequency (brain) signal to understand the gut-brain interaction [23]. To examine EGG-EEG based gut-brain interaction, we utilize the same phase-amplitude coupling measure as the first EGG-MEG study in humans [4]. We hypothesized that EEG would show similar patterns of gut-brain coupling as was previously shown with MEG, and that, further, such gut-brain coupling would be modified by cognitive demands and hunger state.

Prior studies have suggested significant activations in frontal insular area and the occipital area as a part of gastric network based activations [7]. In order to understand whether gut-brain coupling is different within the brain during various cognitive settings and gut satiation, we aim to broadly examine gut-brain coupling in four different scalp electrode clusters that correspond to these brain regions, (1) left fronto-central (median of signals from electrodes F3, C3), (2) right fronto-central (F4, C4), (3) left parieto-occipital (P3, O1), (4) right parieto-occipital areas (P4, O2).

Gut-brain interactions have been studied for their clinical utility and have been shown to be affected in disorders affecting digestion and even mood disorders such as anxiety and depression [1,24,25,26]. In this context, we hypothesize that our study can provide an approach for monitoring with scalable EEG and EGG sensors and characterization of gut-brain interactions during neuro-cognition, which in turn may be used to select clinical intervention strategies.

## 2. Materials and Methods

*Participants.* In this study, we recruited 14 healthy participants from general population (mean ± SD age 25.71 ± 8.32 years, 7 females) with no known gastrointestinal or neurological disorders. Participants were recruited from the local university and city population using flyers. All participants provided written informed consent for the study protocol (#180140) approved by the University of California San Diego institutional review board (UCSD IRB).

*Sample Size and Power*. Our sample size was adequately powered to detect large effect size conditional differences in paired tests (Cohen’s d > 0.81) at beta of 0.8 and alpha significance level of 0.05 as calculated using the G*Power 3.1 software [27].

### 2.1. Procedure and Recording

*Study design.* Each participant completed two visits each on separate days. In every visit, we recorded their electroencephalogram (EEG) and electrogastrogram (EGG) in either hunger or satiety condition, order of condition randomly chosen for every participant. Recordings were made at the NEATLabs in the department of Psychiatry at the University of California San Diego. During each of the visits, the participants engaged with a cognitive suite of assessments for approximately one hour that also included a three minute eyes-closed rest period [11]. For the purposes of this study, analyses of the working memory task and rest data are relevant (more details below). We chose these two cognitive paradigms (working memory, rest) for their importance in the gut-brain interaction literature [19,20]. Participants were requested to fast for at least 8 h prior to their hunger condition session, while they were recorded within 1 h of their latest meal for the satiety condition. All procedures were approved by the UCSD IRB.

*Demographics.* We also obtained demographic variables by self-report including, age, gender, race and ethnicity, socio-economic status measured on the Family Affluence Scale [28], and any current/past history of clinical diagnoses and medications.

*EEG acquisition and processing.* Electroencephalogram (EEG) data was collected for all participants using a 24-channel SMARTING device with a Greentek cap with semi-dry and wireless electrode layout. The electrode sponges were prepared by soaking in saline solution 5–10 min before the recording was started. The cap was fitted tightly on the participants head, with electrodes being placed in the 10–20 system configuration. The schematic of the 24-channel EEG is presented in Figure 1A. We were specifically interested in three specific electrode locations based on their importance from earlier studies [4,29]: left fronto-central (median of signals from C3, F3 electrodes), right fronto-central (C4, F4), left parieto-occipital (O1, P3), right parieto-occipital (O2, P4) electrodes. *EEG Processing.* Data were acquired at 250 Hz sampling frequency at 24-bit resolution. Cognitive event markers were integrated using LSL and data files were stored in xdf format. The xdf file retrieved from the Smarting Streamer app through LSL was analyzed in MATLAB using the EEGLAB toolbox. The EEG signal was first cleaned using the clean_rawdata function in EEGLAB [30], that performs artifact subspace reconstruction for removing bursts, removes flatline channels, and high pass filters the data with transition start and stop band set to [0.25 0.75] Hz. Further, the data were average referenced; only the participants with <10% flatline/missing electrodes (~2 electrodes) identified as bad channels were used in this study; up to 2 noisy/missing channels in any participant were interpolated to nearest neighbors using EEGLAB’s eeg_interp function.

The EEG signals were filtered using Hamming windowed sinc FIR filter (eegfiltnew in EEGLAB) with low cutoff frequency of 9 Hz, high cutoff frequency of 14 Hz to extract the alpha frequency brain rhythm. The filtered EEG signal was Hilbert transformed to derive the instantaneous amplitude. As suggested in prior literature [4], we first calculated the power spectral density (PSD) of the continuous EEG data recorded during both cognitive and resting state of interest in a subject. The dominant power was in the alpha-frequency band (9–14 Hz), and thus further EEG analyses were confined to this frequency.

*EGG acquisition and processing.* The electrogastrogram (EGG) was acquired with 10 disposable cutaneous electrodes (8 active, 1 reference, 1 ground) situated on the abdomen and recorded simultaneously with the EEG. The EGG was configured on the abdomen in a 3 × 3 array with the reference electrode placed to the right of the 2nd row of cutaneous electrodes, as described previously [16]. The schematic of the 10-channel EGG recording, is presented in Figure 1A. Participants were instructed to complete the *BrainE* mobile platform cognitive assessments for about 1 h total engagement. *EGG Processing.* The OpenBCI application was used to process and convert the EGG recording from the microSD file format. The EGG signal was sampled at 250 Hz. To remove non-relevant physiological noise, the EGG signal was fed into a Butterworth bandpass filter with low cutoff frequency of 0.03 Hz, high cutoff frequency of 0.07 Hz [8,17], and filter order equal to 5. This filter range provided a narrow enough passband without distortion for the main EGG frequency of interest, the gastric slow-wave (0.05 Hz). Further artifact rejection was achieved by using a Wiener filter with window size equivalent to the inverse of the dominant frequency of each participant (approximately 20 s) as described previously [16]. The data was normalized by subtracting the mean of the signal from each electrode. The filtered EGG signal was Hilbert transformed and the instantaneous phase was derived. The electrode with the highest power in the [0.03 0.07] Hz frequency range per participant was chosen as the best electrode for phase amplitude coupling analysis for any particular recording session, and the precise frequency with the highest power density was taken as the preferred frequency. Because EGG signals have a 20 s cycle period and each cognitive assessment only takes several minutes to complete, we confirmed the 0.05 Hz peak spectral power over each participants’ entire EGG signal instead of calculating task-specific peak power. The power spectral density (PSD) of the signal was estimated using the Welch method with Hanning window type, segment length equal to the sampling frequency multiplied by 360, and NFFT set to 4 times the segment length. All participants were confirmed to have EGG PSD within acceptable range (0.03–0.07 Hz).

*Working Memory (Lost star).* In the working memory task (Figure 1B), participants accessed a game named Lost Star that is based on the standard visuo-spatial Sternberg task [31]. Participants were presented a set of test objects (blue stars); they were instructed to maintain the visuo-spatial locations of the test objects in working memory for a 3 s delay period, and then responded (yes/no) whether a probe object (green star) had the same location as one of objects in the original test set.

We implemented this task at the threshold perceptual span for each individual, i.e., the number of star stimuli that the individual could correctly encode without any working memory delay [32]. Post-thresholding, the working memory task consisted of 48 trials presented over 2 blocks [33]. Each trial initiated with a central fixation ‘+’ for 500 msec followed by a 1 s presentation of the test set of star objects located at various positions on the screen, then an adaptive working memory delay period, followed by a single probe star object for 1 s, and finally a response time window of up to 1 s in which participants made their yes/no response. A happy/sad face emoticon was used to provide accuracy feedback for 200 msec followed by a 500 msec inter-trial interval. Summary accuracy was also shown between blocks.

The working memory task was performance-adaptive such that all individuals performed the task at ~80% accuracy challenge [34,35]. For this, the initial working memory delay period was set as 1 s. The adaptive paradigm was set in a 3up1down scheme: if correct, the working memory period was increased by +0.9 s; if incorrect, the memory period was decreased by −0.3 s with minimum at 1 s (i.e., no decrease beyond initial memory period of 1 s). Max trials were based on 10 trial accuracy reversals (20 trials batch shuffled at a time) or all correct up to 40 trials. The total run length was ~10 min.

As behavioral accuracy was thresholded across subjects, we analyzed behavioral data for performance speed, where speed = log(1/RT), and RT is response time in milliseconds.

*Resting state.* During the resting state, the participants closed their eyes and rested for about 3 min.

### 2.2. Brain-Gut Coupling Analyses

Time synchronization between the simultaneously collected data streams from the electrogastrogram and electroencephalogram were performed using time locked markers from lab streaming layer, LSL [36]. The EEG and EGG signal segments corresponding to the working memory task and resting were extracted from the full recording [11]. For computing the brain-gut coupling, we did not time lock the EEG and EGG data from cognitive and resting state to any triggers, and thereby use continuous signals for our below analysis.

*Phase-Amplitude Coupling (PAC).* The instantaneous amplitude of the fast frequency EEG signal for phase-amplitude coupling analysis was evaluated by taking the absolute value of the Hilbert transform of the filtered EEG signal. The instantaneous phase of the slow frequency EGG signal for phase-amplitude coupling analysis was evaluated by taking the angle of the Hilbert transform of the filtered EGG signal. The direct PAC estimator (Equation (1)) was computed based on theory published previously [23].
(1)$$PAC\left(f_{P},f_{A}\right)=\frac{1}{\sqrt{T}}\frac{\left|\sum_{t=0}^{T}A_{f_{A}}\left(t\right)\cdot e^{i\Phi_{f_{P}}\left(t\right)}\right|}{\sqrt{\sum_{t=0}^{T}A_{f_{A}}{\left(t\right)}^{2}}}$$
where *Φ_fp_*(*t*) is the instantaneous phase of the low-frequency oscillation computed in MATLAB using angle() function of the hilbert transformed EGG best-electrode data, and *A_fA_*(*t*) is the instantaneous amplitude of the high frequency oscillation computed using abs() function of the hilbert transformed EEG data.

The coupling was measured using the direct mean vector length of the analytical signal, which is the data point whose amplitude was extracted from the high frequency EEG and phase from the low frequency EGG. Each complex value of the analytical signal was a vector; averaging all vectors generated the mean vector with a specific phase and length, which represented the mean phase where the amplitude was strongest, and the magnitude of PAC, respectively (see Figure 2). In particular, this exercise was carried out for the portion of signals whose analytical amplitude was in the top 5% in any instance. This measure was further normalized to the [0 1] range to take care of possible differences in amplitude in raw data. The PAC calculation we used has been shown to be sensitive and specific to true underlying PAC detection similar to other commonly used measures such as modulation indices [37,38,39]. In addition to the normalization, the PAC values were log-transformed for all subsequent analyses.

*Surrogate Control Analysis.* To determine if the measured PAC is resultant of true physiological gut-brain coupling or random signal fluctuations, we performed random permutation analysis [40]. The amplitude vector of a participant was randomly picked for any cognitive state (i.e., working memory or rest) or gut state (i.e., hunger or satiety) and EEG electrode and shuffled so any correspondence is broken before computing PAC with the EGG phase vector, thus breaking any existing phase-amplitude relationships between the two signals while preserving physiological variance. The PAC was then calculated for the shuffled EEG amplitudes and EGG phases.

3000 such phase-shuffled surrogate PACs were calculated for each electrode and each gut state (i.e., hunger/satiety) and cognitive state (i.e., working memory or rest), to produce a chance distribution, and the threshold of significant coupling was set as the top 5% of these surrogate PAC values. Any experimental PAC value (PAC with non-shuffled EGG phase) observed above the threshold was considered to be resultant of true physiological gut-brain coupling.

### 2.3. Statistical Analysis

Paired *t*-tests were used to analyze the mean PAC values in each electrode across participants, in comparison to their random control values. PAC in different brain areas were analyzed using robust linear regression implemented with robust linear fit function in MATLAB [41,42,43,44] with cognitive state (rest/working memory) and gut state (hunger/satiety) as two categorical covariates along with their interaction. We used FDR corrections for multiple comparisons. Cohen’s d effect sizes are reported for all comparisons (d = 0.2 is small effect size, 0.5 = medium, 0.8 = large). PAC measures were also analyzed using repeated measures analyses of variance (rm-ANOVA, implemented using fitrm command in MATLAB) with within-subject factors of brain regions (left fronto-central (median of signals from electrodes-F3, C3; right fronto-central-F4, C4; left parieto-occipital-P3, O1; right parieto-occipital areas-P4, O2), gut state (satiety, hunger) and cognitive state (rest, working memory); the Tukey–Kramer method was used for post hoc testing.

Spearman correlations were used to assess monotonic relationships between the PAC and the task performance, specifically working memory speed across participants. FWER tests were applied to correct for multiple comparisons across cognitive/gut states and four brain regions. A linear least squares regression solver was used to plot a line of best fit for significantly correlated data points.

## 3. Results

We performed simultaneous EEG and EGG recording in 14 participants across two visits on separate days. All participants reported normal/corrected-to-normal vision and hearing and no participant reported color blindness. All of the participants were right handed. In each of the visits, the participants were either in a state of hunger or satiety with order of hungry day randomized across participants. During each of the visits, the participants engaged with a cognitive suite of assessments that also included a three minute eyes-closed rest period. For the purposes of this study, analyses of the working memory task and rest data are relevant. The schematic of the 24-channel EEG, 10-channel EGG recording, and the working memory task design is presented in Figure 1A,B.

Consistent with prior literature [4], the dominant continuous-EEG power was found in the alpha-frequency band (9–14 Hz), and thus further EEG analyses were confined to this frequency (Figure 3A). We next calculated the normalized log-transformed Phase-Amplitude Coupling (PAC, see Methods [23]) obtained from low-frequency EGG activity (0.03–0.07 Hz) phase and EEG alpha (9–14 Hz) amplitude for each EEG electrode. Exemplars of PAC estimates across cognitive states (working memory vs. rest) and satiety states (hunger vs. satiety) suggest diversity at the level of individuals, and relatively higher coupling during resting state (Figure 3B). Topographic plots of the alpha band PSD and PAC are shown in Figure 3C, these initial topographies are only shown for resting state as prior research has only focused on resting state gut-brain coupling [4]. Topographic maps were FDR corrected for multiple comparisons across electrodes for finding t-test significance against random permutation null distributions in all electrodes; all electrodes survived the statistical corrections. After averaging across hunger and satiety conditions in all participants during resting state, the alpha-PSD plot shows maximal PSD in the midline parieto-occipital brain region (Figure 3C(i)) consistent with those previously reported from MEG-EGG recordings, PAC measures also suggest relatively higher coupling in fronto-central and parieto-occipital areas (Figure 3C(ii)), again consistent with earlier MEG-EGG results [4].

### 3.1. Permutation Control Analysis Reveal Significant Gut-Brain Coupling by Concurrent EEG-EGG

We computed normalized log-transformed PAC for resting and cognitive states between the dominant alpha brain rhythm (9–14 Hz, Figure 3) in each brain electrode and the gut rhythm signals from the best gut electrode (0.03–0.07 Hz, see Methods section for selection of best electrode). We generated random permutation control distributions to verify that the PAC values obtained for each electrode and condition were not due to chance or were affected by the small sample size and the signal lengths used for computing the PAC (see Methods on Surrogate control). PAC comparisons against the surrogate control in every electrode were significantly different (paired ttest, tstat(13) = 4.60, *p* = 0.0005). The PAC statistics against control permutations were FDR corrected for multiple comparisons across factors including individual electrodes (24 electrodes), resting and working memory cognitive states (2 cognitive states), and hunger and satiety states (2 gut states); these are shown in Figure 4 hunger and satiety topographic maps (non-significant PAC values are programmed to take value = 0 with color green) and all electrodes survived multiple comparison corrections.

During both hunger and satiety states, in both cognitive conditions, we found the strongest PAC in the fronto-central and occipital areas. Significant differences in the PAC between hunger and satiety states were largest in the central area at rest and in frontal regions during the working memory task (Figure 4). Moreover, PAC topographies did not completely overlap with the alpha PSD topographies, averaged for hunger and satiety states, shown in the last row of Figure 4. For instance, the alpha PSD did not show any significant differences between hunger and satiety whereas substantial differences were seen with PAC activations.

### 3.2. Gut-Brain Coupling Differs with Cognitive Engagement and Hunger State

Prior studies have only examined gut-brain coupling during resting state. Here, we uniquely analyzed PAC between the brain and gut across both resting and working memory cognitive states and different satiety states within the same participants. Our earlier full brain (24 channel EEG-EGG) measurements suggest they are similar to those found in other studies using MEG-EGG. We next ask two questions (1) whether there are any significant differences between gut state of hunger/satiety and cognitive state of rest/working memory within different brain regions, and (2) whether individual differences in gut-brain coupling correlate with cognitive performance.

For these analyses, we compared PAC by gut state and cognitive state in four different brain regions: (1) left fronto-central (median of signals from electrodes F3, C3), (2) right fronto-central (F4, C4), (3) left parieto-occipital (P3, O1), (4) right parieto-occipital areas (P4, O2), to simplify our analysis in correspondence to the observed topographical contrasts (Figure 4).

We then applied robust linear regression to analyze log-transformed PAC effects as the single response variable with within participant factors of cognitive state (resting = 0, working memory = 1) and gut state (hunger = 0, satiety = 1) modeled as categorical binary covariates [41,42,43,44]. The interaction between satiety condition and cognitive condition was used as an additional categorical covariate. Results are shown in Table 1. We found that all four regression models were significant with FDR corrections applied for 4 different brain regions, 4 different cognitive and gut states for left/right fronto-central (left: R^2^ = 0.27, df = 52, *p* = 0.002; right: R^2^ = 0.17, df = 52, *p* = 0.02) and left/right parieto-occipital (left: R^2^ = 0.21, df = 52, *p* = 0.007; right: R^2^ = 0.18, df = 52, *p* = 0.02) sites. Further, there were significant cognitive state dependent PAC modulations in left/right fronto-central and left parieto-occipital sites (Table 1). In addition, in the left fronto-central region, there was significant hunger vs. satiety gut state-based modulation of PAC (*p* = 0.03, Table 1) as well as significant interaction of satiety state with cognitive state (*p* = 0.003, Table 1).

We further conducted rm-ANOVAs on the PAC values with the brain regions (left, right fronto-central, left, right parieto-occipital), gut states (satiety, hunger) and cognitive states (rest, working memory) as within-subject factors, to investigate whether hunger/satiety effects on PAC were significantly different from each other across the two cognitive states and brain regions. We found a significant main effect of brain regions (Fstat(39) = 3.26, *p* = 0.03), cognitive state (Fstat(13) = 8.38, *p* = 0.01), and a significant interaction effect between brain regions, gut states and cognitive states (Fstat(39) = 3.88, *p* = 0.016). Subsequently, we conducted rm-ANOVA in each of the 4 brain regions (Table 2). These analyses showed a main effect of cognition in the bilateral fronto-central regions, and in the left parieto-occipital regions, with greater PAC observed in resting state relative to the working memory task. Further, we found a significant interaction effect between the gut and cognitive states in the left fronto-central region (see table below); post hoc tests in this region revealed significant PAC differences only in the hungry state with greater PAC observed during resting state vs. during working memory (*p* < 0.03).

Figure 5 shows a summary of PAC (median and standard error of median) in the four brain regions during the two cognitive and two gut states. We observed that overall PAC during rest exceeded PAC during working memory in all four brain regions. Further, the extent to which this PAC modulation occurred depended on the brain region as well as satiety level. In particular, the smallest change in PAC from rest to working memory occurred in the satiety state in the left fronto-central region (ttest, tstat(27) = 0.99, *p* = 0.22, Cohen’s d = 0.27) and the largest change occurred in the hungry state in the same region (tstat(27) = 3.19, *p* = 0.002, Cohen’s d = 0.85).

### 3.3. Gut-Brain Coupling Relates to Behavioral Performance Speed

We next considered whether individual differences in gut-brain coupling relate to behavioral performance on a working memory task. In particular, we measured the behavioral speed or 1/response time in our adaptive behavioral task (wherein performance accuracy is adaptively controlled) as a function of gut-brain PAC. We found there is a significant relationship to working memory speed only in the left parieto-occipital region (Spearman r = 0.60, *p* = 0.002, FWER corrected for multiple comparisons across 4 brain regions, Figure 6, blue) after partially controlling for the satiety level; no such correlations were found at rest (Figure 6, red).

## 4. Discussion

In this study, we explored how a manipulation of the gut state, i.e., hunger or satiety, as well cognitive state, i.e., resting state versus working memory, affects the levels of gut-brain coupling. We used EGG and EEG to record electrical activity from the gut and the brain simultaneously, and employed a normalized measure for phase-amplitude coupling between the visceral gut signal and the brain’s dominant frequency alpha rhythm. The PAC measure we used has been shown to circumvent some of the amplitude and signal computation confounds in the quantification of coupling measures [23].

We observed PAC using EEG-EGG consistent with earlier observations using MEG-EGG [4]. PAC in the fronto-central and parieto-occipital areas during resting state were relatively enhanced compared engagement during the working memory task. More specifically, in the left fronto-central region, we observed significant PAC differences only in the hungry state (*p* < 0.03).

The function of the gastric network is suggested to be linked to food intake, as evidenced by gastric-BOLD (blood-oxygen-level-dependent) coupling being particularly salient in the hand and mouth mapping brain regions [8]. Activity in the insula in the right frontal region and the occipital cortex (also known as visual cortex) have been shown to be significant parts of the neural gastric network. While the insula’s involvement in the gastric network can be understood given that the region serves as the ‘primary visceral cortex’ [8], the involvement of the occipital cortex in the gastric network is less clear [7]. Rebollo et al. postulate that the role of the occipital cortex in the gastric network is related to vigilance as part of a homeostatic regulation mechanism underlying feeding. The basic action of finding food requires one to be able to identify a food source, which may translate to being visually vigilant for feeding opportunities.

Our study explores the differential effect of cognitive and gut states on gut-brain coupling. For simplicity, we focused on four brain regions: left/right fronto-central, left/right parieto-occipital areas. In linear regression analyses, we found that all four brain areas showed significant PAC effects for the type of cognitive engagement, with increased coupling during resting state than in the working memory state, in both hungry and satiety conditions. Notably, the left fronto-central region also showed significant interaction between cognitive state and gut state on PAC. Specifically, during hunger, there was relatively increased coupling in the resting state than in the cognitively engaged working memory state. No such differences were found during satiety between cognitive states.

Many functional imaging studies have demonstrated the distinct roles of brain regions in cognitive control. In particular, the left fronto-central site, corresponding to the left dorsolateral prefrontal cortex (dlPFC), has been the widely recommended target for brain stimulation in severe depression [45,46]. Our study suggests the potential for gut state as an effective modulator for tuning the activations of the dlPFC region and therefore contribute to more precise neuropsychiatric interventional strategies [47,48].

Overall, we found the gut-brain coupling to be stronger during resting state than in the cognitively engaged state. One plausible explanation is the “rest and the digest” hypothesis, where the gut actions are highly active during the resting state predominantly due to parasympathetic activations [5,7].

Interestingly, the PAC between the brain and the gut also correlated to behavioral performance speed on the working memory task, especially in the left parieto-occipital region. We observed increased working memory response speed with increased gut-brain coupling in this area. In accordance with the hypothesis proposed by Tallon-Baudry et al., increased coupling in this area correlates with increased vigilance, which may underlie the enhanced cognitive performance speed that we observed [6,7].

The major limitation of this first study of EEG-EGG coupling is its small sample size such that results need replication in future large sample work, ideally with more dense electrode montages. Altogether, this set of first results from simultaneous EEG-EGG recordings nonetheless leads the way to a new domain of inquiry for understanding the effect of cognitive demand on gut-brain interactions. Since the study uses scalable and portable sensor recording techniques from the brain and the gut, there is potential for many research groups to participate in this newly emerging research area and further contribute to clinical applications that rely on gut-brain interactions.

## Figures and Tables

**Figure 1 sensors-22-09242-f001:**
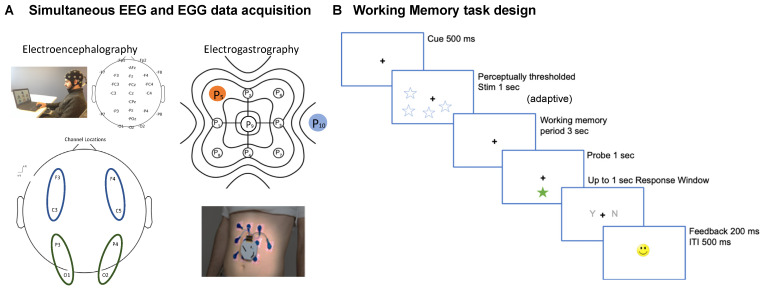
**Procedure schematic.** (**A**) Recording setup and electrode layout used for our 24-channel electroencephalography (EEG), highlighted with the 4 different regions (each 2 electrodes, left fronto-central (median of signals from electrodes F3, C3), right fronto-central (F4, C4), left parieto-occipital (P3, O1), right parieto-occipital areas (P4, O2) used for study analysis), and 8-channel electrogastrography (EGG, red: reference, blue: ground) is shown. (**B**) Task design for the working memory task is shown.

**Figure 2 sensors-22-09242-f002:**
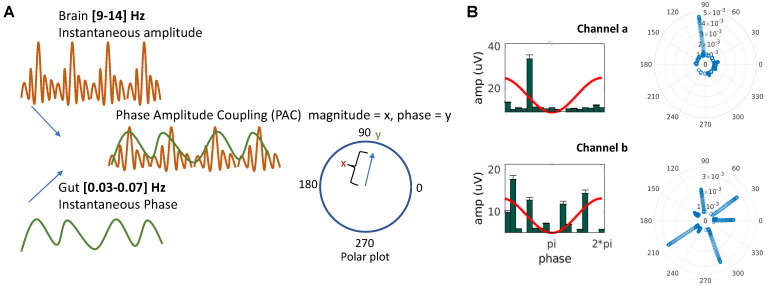
**Phase Amplitude Coupling (PAC).** (**A**) Illustration for computing PAC with instantaneous amplitude of the fast frequency EEG signal and instantaneous phase of the slow frequency EGG signal, and polar plot representation of the resultant coupling mean vector magnitude (x) at phase (y) (**B**) Illustrative examples of high coupling at phase ~100 degrees (top), and dispersed coupling spread over various phases (bottom).

**Figure 3 sensors-22-09242-f003:**
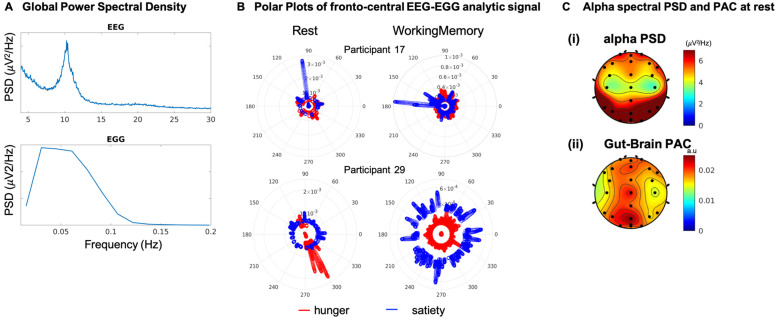
**Procedure schematic.** (**A**) Global power spectral density (PSD) across all participant data showed a dominant EEG rhythm peaking in the alpha band (9–14 Hz) and dominant EGG rhythm peaking in the 0.03–0.07 Hz range. (**B**) Phase and normalized magnitude of the EEG-EGG analytical signal shown as polar plots (see Methods Figure 2) for two exemplar participants in the left fronto-central brain region (F3, C3), these participants data suggest diversity in brain-gut coupling across conditions (levels of satiety, cognition) at the level of individual participants. (**C**) The top plot (i) represents the normalized alpha band PSD in uV2/Hz. The bottom plot (ii) represents normalized and log-transformed PAC showing midline areas with elevated PAC. All signals survived FDR statistical corrections for multiple comparisons across electrodes. PAC: phase amplitude coupling.

**Figure 4 sensors-22-09242-f004:**
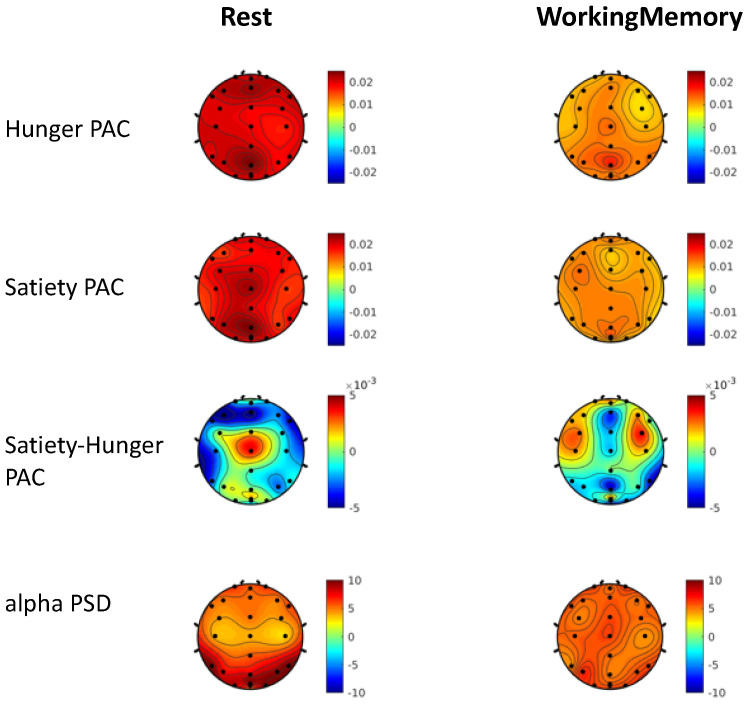
**PAC during cognitive and gut states.** PAC values against permutation control distributions were FDR corrected, and are shown for eyes closed resting state and working memory cognitive state during hunger, satiety, and satiety-hunger contrast. Last, we present the spectral power density averaged for hunger and satiety states which showed increased density in the parieto-occipital area. There was not any significant difference between hunger and satiety in alpha PSD, in contrast to the significant PAC difference observed in frontal and central areas.

**Figure 5 sensors-22-09242-f005:**
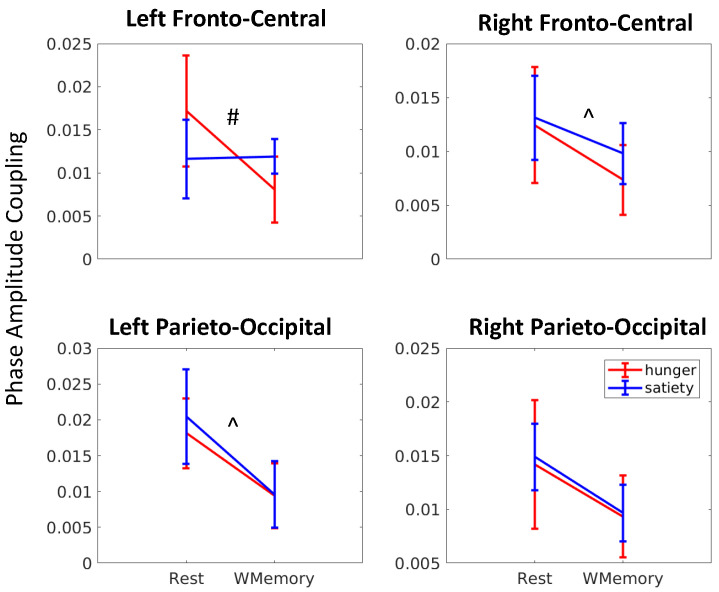
**Phase amplitude gut-brain coupling in four brain regions differs by cognitive state and gut satiety state.** Significant cognitive state (resting versus working memory) based differences in PAC (marked with ^) are observed in the left fronto-central (C3, F3), right fronto-central (C4, F4), and left parieto-occipital (P3, O1) regions. Modulations in the right parieto-occipital (P4, O2) region were not significant. Specifically, the left fronto-central region also showed significant (marked with #) hunger vs. satiety gut state effects, and significant cognitive state differences that vary with the levels of hunger (refer to Table 1 for statistical results).

**Figure 6 sensors-22-09242-f006:**
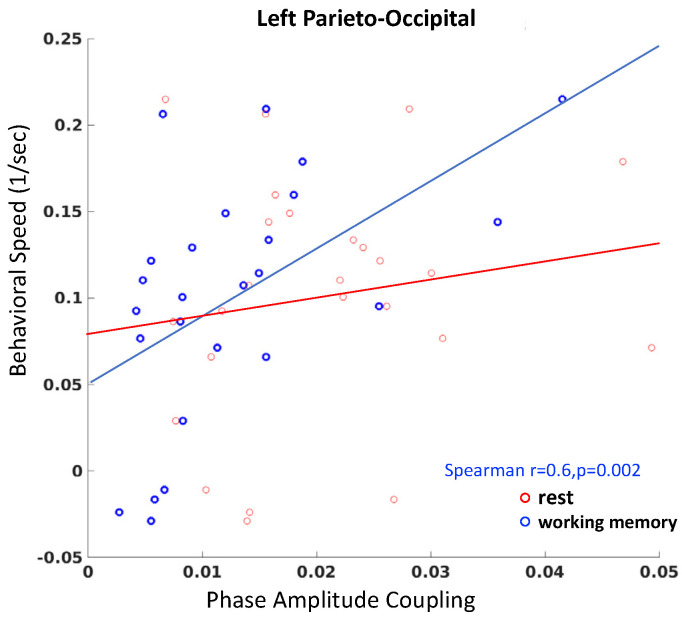
**Spearman correlations between PAC and behavior.** A significant relationship was found between PAC during the working memory task and performance speed, with increased coupling corresponding to enhanced performance speed (r = 0.6, *p* = 0.002).

**Table 1 sensors-22-09242-t001:** Robust linear regression effects of satiety and cognitive state on PAC. The table summarizes the regression weights, tstat and the *p*-value, for the coefficients denoted by the factor column for each of the electrode models. Significant *p*-values less than 0.05 are in bold.

Brain Regions	Factor	Robust Linear Fit Estimate ± SE	Robust Linear Fit tstat	Robust Linear Fit *p*-Value
Left fronto-central	Satiety	−0.11 × 10^−2^ ± 0.5 × 10^−3^	−2.19	**0.03**
	Cognition	−0.20 × 10^−2^ ± 0.5 × 10^−3^	−3.90	**0.0003**
	interaction	0.16 × 10^−2^ ± 0.7 × 10^−3^	2.19	**0.03**
Right fronto-central	Satiety	−0.15 × 10^−3^ ± 0.4 × 10^−3^	−0.32	0.74
	Cognition	−0.10 × 10^−2^ ± 0.4 × 10^−3^	−2.21	**0.03**
	interaction	0.56 × 10^−3^ ± 0.6 × 10^−3^	0.83	0.41
Left parieto-occipital	Satiety	0.44 × 10^−3^ ± 0.7 × 10^−3^	0.63	0.52
	Cognition	−0.14 × 10^−2^ ± 0.7 × 10^−3^	−2.05	**0.04**
	interaction	−0.26 × 10^−3^ ± 0.9 × 10^−3^	−0.26	0.79
Right parieto-occipital	Satiety	−0.36 × 10^−3^ ± 0.6 × 10^−3^	0.58	0.56
	Cognition	−0.80 × 10^−3^ ± 0.6 × 10^−3^	−1.29	0.20
	interaction	0.23 × 10^−3^ ± 0.8 × 10^−3^	−0.27	0.78

**Table 2 sensors-22-09242-t002:** Repeated measures ANOVA effects of satiety and cognitive state on PAC. The table summarizes the ANOVA Fstat and *p*-value, for the main effects and interactions for each of the brain regions. Significant *p*-values less than 0.05 are in bold.

Brain Regions	Main Effects and Interaction	rm-ANOVA (df = 13) Fstat	rm-ANOVA (df = 13) *p*-Value
Left fronto-central	Satiety	1.48	0.25
	Cognition	9.11	**0.009**
	interaction	7.21	**0.02**
Right fronto-central	Satiety	0.04	0.84
	Cognition	7.39	**0.02**
	interaction	0.26	0.61
Left parieto-occipital	Satiety	0.10	0.76
	Cognition	8.65	**0.01**
	interaction	0.45	0.51
Right parieto-occipital	Satiety	0.03	0.87
	Cognition	3.98	0.07
	interaction	2.32	0.15

## Data Availability

The data will be shared on specific request to the authors.

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
