# Peer review of "Simultaneous Gut-Brain Electrophysiology Shows Cognition and Satiety Specific Coupling"

_sensors, 2022, doi:10.3390/s22239242_

Round 1

Reviewer 1 Report

I am okay with the current version of manuscript.

Author Response

We are thankful and encouraged by the reviewer’s positive decision.  

Reviewer 2 Report

The authors proposed an original technique to assess the effect of satiety factor on brain activity, including in the cognitive task. The results can attract much attention, and this is of great value and interest to readers. However, I would like to write a couple of comments:

Comments 1: Unfortunately, the conducted statistical analysis does not give an overall view and a holistic picture of the research. However, a multivariate analysis of variance with repeated measures can provide the most complete representation of the relationship of all factors and their influence on the estimated variables. I earnestly ask the authors to perform MANOVA, so this will significantly improve the quality and reliability of the results. The relatively small number of participants and short EEG recordings need additional variance comparisons.

Comments 2: Please in the section of methods specifies the total number of EEG epochs, as well as within each state. Are the samples comparable in the number of examples?

Comments 3: The conclusions section of the paper is missing. I ask the authors to add this mandatory section.

I will be glad to read the article again after its major revision.

Author Response

We are thankful for the reviewer’s encouraging comment, and thoughtful suggestions for improving our manuscript. We have made several updates to our manuscript based on the reviewer’s recommendations. Please see the attachment.

Reviewer 3 Report

Balasubramani and collaborators looked at the gut-brain coupling using EEG and EGG under hunger and satiety conditions during a cognitive task and at rest. The main observation is that the gut-brain coupling to be stronger during resting state than in the cognitively engaged state. Furthermore, in the left fronto-central region, during hunger, coupling is stronger in the resting state than in the cognitively engaged  state.

Abstract:

"We find that EGG-EEG phase-amplitude coupling differs based on both satiety state and cognitive effort." Please qualify those differences.

"these results highlight the important role of gut-brain interactions in cognition" The results highlight a role of the gut-brain coupling, but I'm not sure that the 'importance' of it is clearly highlighted. Consider using a closing sentence that might enlighten the reader as to what the clinical significance of the results might be.

Introduction:

This section is generally well written and introduces the rationale quite well. However, the clinical relevance of the study is missing. How might the results obtained be translated in clinical practice? An hypothesis in relation to working memory is also missing. What are we expecting to see in terms of the gut-brain interaction during the working memory task and how do we think the state (hunger/satiety) might influence the signal and/or performance?

Materials ad Methods:

In opposition to the introduction, the methods section presents several issues which make it very difficult to read and understand.

  • First sentence: please replace 'analyzed' by 'recruited'
  • Please replace 'subject' by 'participant' throughout the text
  • Were participants recruited from the general population?
  • Information obtained from the participants should be part of the procedure, not part of the 'participants' section. Also, if normal vision and hearing, as well as handedness were a inclusion criteria, please make this clear. If they were not inclusion criteria, then this information should be presented in the results section.
  • The section paragraph should also be part of the procedure under the subtitle 'study design'
  • "Each participant made two visits." Please replace 'made' by 'completed'
  • The hunger condition is described, but not the satiety condition.
  • When introducing BrainE, in the design section, it should already be specified that only the working memory task of this cognitive battery is used to avoid confusion.
  • To the 'participants' section, please specified whether the study was ethically approved and whether participant signed a consent form.
  • Sample size, please show the actual calculations
  • Page 3, line 107: replace 'inserted' by 'placed'
  • P3, line 122, resting state is not a cognitive task.
  • Could the lost start description be simplified? It's quite long and somewhat confusing.
  • 'EEG processing' section: information about acquisition should all be presented together. I suggest reordering the whole Procedure & Recording section the following way: Demographics data, EEG acquisition and processing, EGG acquisition and processing, Working Memory task, resting EEG, PAC
  • P4, line 178, the rational for the focus on Alpha, as presented in the results section, should be presented here. Also, which EEG signal was used to calculate the PSD?
  • The PAC section is very hard to understand and I would suggest simplifying the text to help a reader who may not be familiar with this type of analyses. I understand that simplifying will be challenging. A figure may help? Indeed, it would be useful to visualise those 'vectors'. Also, after reading the methodology section, I can't say that I understand how the analyses are time-locked with the memory task triggers. I also don't understand how the resting state data is analysed.
  • It seems that the EEG acquisition system is different between p.3 lines 105-107 and p. 4 lines 163-165
  • Statistical analysis: "PAC values across all conditions". I'm confused as to what all the conditions are.
  • From what I could understand from the PAC section is that this coupling method bring together both EEG and EGG data into a single variable. It is therefore unclear to me why EEG channels are analysed. And what variable from the EEG channel is analysed?
  • I also question the use of paired t-tests for the PAC values analyses, shouldn't mixed effect models  be used?

Results:

  • The first paragraph is actually a really good to explain the design of the study, This information does is not a ';result' and I wish I had found this paragraph in the methods section! Figures 1A and B should also definitely be part of the methods.
  • Figure 1D results: looping back to my comment around the PAC analysis and how a figure in the methods would be useful to better understand. I assume that what I'm looking at here are the so-called 'vectors'? What does a long or a short vector means? What does the orientation tell me? This is the kind of information I would like to see in the methods.
  • Figure 1D: why were those two participants chosen?
  • Please indicate what the legend for the topography plots indicates.
  • I'm not sure what Figures 1E i and ii are meant to show us. Are they needed?
  • The results section could be shortened by removing all the methodological repetitions. I'm the methods section was clear, there would be no need to repeat the methods when presenting the results.
  • Line 341, Figure 3 should not be introduced here.
  • "To simplify our analysis, corresponding to the topography contrasts between satiety and hunger

states during resting state versus working memory" Please clarify, the same analyses as before are conducted again, but this time on a per region basis? If I'm wrong, then please improve the manuscript so the reader can understand better. If I'm right, then I'm not sure how this new analysis is 'simplifying'? And if the aim was to present simplified analyses, then why present the full brain ones? The rationale for the different analyses used need to be better explained. Not necessarily in technical word, but in terms of their neurphysiological meaning.

Discussion:

  • "We observed relatively enhanced PAC" Relative to what?
  • "with the difference between satiety and hunger being maximal" In what direction does this difference go?
  • "While in the cognitively engaged state, we observed increased PAC" Relative to what? Resting state?
  • You need to chew the information for the reader. I don't want to have to search for it.
  • Line 417, please delete the first "which receives visceral inputs"
  • In the occipital cortex explanation, please clarify that this region is also known as the visual cortex.
  • The fourth paragraph has enabled me to understand the results that I was still unclear with after reading the whole results section.
  • "understanding gut-brain interactions and their effects on cognition and health." This study was rather looking at the effect of cognitive demand on the gut-brain interaction.

Author Response

We thank the reviewer for their constructive suggestions. We have made several updates to our manuscript based on the reviewer’s recommendations. Please see the attachment.

Round 2

Reviewer 2 Report

Many thanks to the authors for the major revision of the manuscript. All my comments received satisfactory answers. I recommend the article for publication. 

Reviewer 3 Report

I wish to thank the authors for addressing my comments to satisfaction. The manuscript, especially the methods sections is much improved.